# ANTMAN: SPARSE LOW-RANK COMPRESSION TO ACCELERATE RNN INFERENCE

## ABSTRACT

Wide adoption of complex RNN based models is hindered by their inference performance, cost and memory requirements. To address this issue, we develop AntMan, combining structured sparsity with low-rank decomposition synergistically, to reduce model computation, size and execution time of RNNs while attaining desired accuracy. AntMan extends knowledge distillation based training to learn the compressed models efficiently. Our evaluation shows that AntMan offers up to 100x computation reduction with less than 1pt accuracy drop for language and machine reading comprehension models. Our evaluation also shows that for a given accuracy target, AntMan produces 5x smaller models than the state-of-art. Lastly, we show that AntMan offers super-linear speed gains compared to theoretical speedup, demonstrating its practical value on commodity hardware.

## 1 INTRODUCTION

Remarkable advances in deep learning (DL) have produced great models across a wide variety of tasks such as computer vision, machine reading, speech generation and image recognition (Goodfellow et al., 2016). However, wide adoption of these models is still limited by their inference performance, cost and memory requirements. On the client side, all pervasive devices like smart-phones, tablets and laptops have limited memory and computational resources to handle large complex DL models. On the server side, intensive computation can render the models too slow to meet responsiveness requirements and too expensive to scale, preventing their deployment in production.

Model Compression is a flourishing area that aims to reduce the computational and memory complexity of DL models to address the aforementioned problems without significantly affecting accuracy. Compressing Convolution Neural Networks (CNNs) have already been widely explored in the past few years (Cheng et al., 2017), while our work focuses on Recurrent Neural Networks (RNNs), which are broadly used among various natural language processing tasks (Mikolov et al., 2010; Seo et al., 2016; Zaremba et al., 2014). It is well known that large RNN models are computation and memory intensive (Zhang et al.). In particular, their computation increases linearly with sequence length, and their recurrent unit has to be computed sequentially, one step at a time with limited parallelism, both of which makes long execution time a crucial issue for RNN inference computation. Compressing RNNs, however, is challenging, because a recurrent unit is shared across all the time steps in sequence, compressing the unit will aggressively affect all the steps.

Inducing sparsity is one of the prominent approaches used for RNN compression. Narang et al. (2017a) proposed a pruning approach that deletes up to 90% connections in RNNs. The obtained sparse matrices, however, have an irregular/non-structured pattern of non-zero weights, which is unfriendly for efficient computation in modern hardware systems (Lebedev & Lempitsky, 2016; Wen et al., 2016). To address this issue, Narang et al. (2017b) proposed inducing block-sparsity in RNNs via pruning or group lasso regularization. Similarly, Wen et al. (2017) introduces ISS, intrinsic structured sparsity for LSTMs (Hochreiter & Schmidhuber, 1997), a type of RNN , such that a sparse LSTM can be transformed into a dense one but with smaller size. ISS conveniently turns sparsity into efficient execution, but as its sparse structure is quite coarse-grained, it is hard to push out high sparsity without degrading accuracy, especially in RNNs where the hidden dimension is smaller than input dimension (elaborated in Section 5.1).

Our work explores a new line of structured sparsity on RNNs, using predefined compact structures as opposed to pruning and regularization based approaches. We take inspiration from predefined compact CNN structures such as group convolutions (Zhang et al., 2017; Krizhevsky et al., 2012)

and depth-wise separable convolutions (Chollet, 2017). Specifically, we replace matrix-vector multi-plications (MVs), the dominant part of RNN computations, with localized group projections (LGP). LGP divides the input and output vectors into groups where the elements of the output group is computed as a linear combination of those from the corresponding input group. In addition, to em-power the information flow across multiple groups along the steps of RNN computation, we use a permutation matrix or a dense-square matrix to combine outputs across groups, helping the compact structure to retain accuracy.

Furthermore, we combine LGP with low-rank matrix decomposition in order to further reduce the computations. This is possible as low rank and sparsity are complimentary to each other. Low-rank decomposition such as SVD approximates a low-rank multiplication $A\mathbf{x}$ as $PQ\mathbf{x}$, where $P$ and $Q$ are dense. By imposing LGP-based sparsity on $P$ and $Q$, we reduce the computation further. For a given rank reduction factor of $r$, we reduce the computation cost and model size by $\mathcal{O}(r^2)$, compared to $\mathcal{O}(r)$ by using low-rank decomposition methods like SVD (Golub & Reinsch, 1970) alone.

We call our compression approach AntMan — 'shrink in scale' by synergistically combining struc-tured sparsity and low-rank decomposition, but 'increase in strength' by enabling the flow across structured groups along RNN sequence to retain accuracy.

To train RNN models with AntMan, we use teacher-student training paradigm (Bucilu et al., 2006) by combining the label loss with teacher-MSE-loss and teacher-KL-divergence-loss. To improve the training efficiency, we develop a new technique to decide proper coefficients to obtain high accuracy efficiently with minimal trials.

We evaluate AntMan on multiple RNN based models for machine reading comprehension and lan-guage modeling. For a well-known MRC model (Seo et al., 2016), we reduce the computational complexity and model size of LSTMs (a particular type of RNN) by up to 25x with less than 1pt drop in F1 score. For PTB (Zaremba et al., 2014) language model, we achieve a computational reduction of 50x with no drop in perplexity, and 100x with just a single point drop in perplexity. We also construct language models for PTB with perplexities ranging from 64 to 70, but with 3x to 5x fewer overall model weights (5x to 25x reduction in RNN weights) than the state-of-art.

Last but not least, we develop efficient implementations of inference kernels on CPUs to serve mod-els compressed by AntMan. We show that unlike computation with unstructured sparsity, AntMan offers significant performance improvement for large RNN models even with modest levels of spar-sity. Our evaluations show that a 2x to 10x theoretical reduction in computation can result in up to 2x to 30x actual speedup, respectively, for moderate to large RNNs, demonstrating attractive practical value of AntMan on commodity hardware.

## 2 RELATED WORK

**Compressing RNNs via Sparsity:** Described in Section 1 and empirically compared in Section 5.1.

**Compressing RNNs via Low-Rank Approximations:** Prabhavalkar et al. (2016); Lu et al. (2016) use SVD to compress LSTM models by 3-4x for acoustic modeling and speech recognition tasks with negligible loss in accuracy. AntMan achieves significantly higher compression rate than SVD based methods for the same rank reduction. Ye et al. (2017) uses Block Tensor Decomposition to compress LSTMs for vision tasks. Their work is specifically designed to exploit redundan-cies present in the image vector (input to the LSTMs) obtained from upstream CNN layers, while AntMan is designed to compress general RNNs, where the inputs do not exhibit such redundancies in many cases.

**Teacher-Student training paradigm:** Knowledge Distillation (KD) technique developed by Hinton et al. (2015) is a popular approach to compress deep and wide networks into sparser ones, where the compressed model mimics the function learned by the complex model. KD usually optimizes a weighted average of *two* different objective functions. The first objective function can be one of the following three: cross entropy, or mean square error, or Kullerback Leiber divergence, all computed with respect to the soft targets, and the second objective function is the cross entropy with the correct labels. Several similar approaches (Romero et al., 2014; Luo et al., 2016; Chen et al., 2015; Balan et al., 2015; Zagoruyko & Komodakis, 2016) extend the idea of KD.

In contrast, AntMan optimally combines *three* objective functions, MSE loss, KL divergence loss and the cross entropy of the true labels, powered by an efficient method of deciding their coefficients.

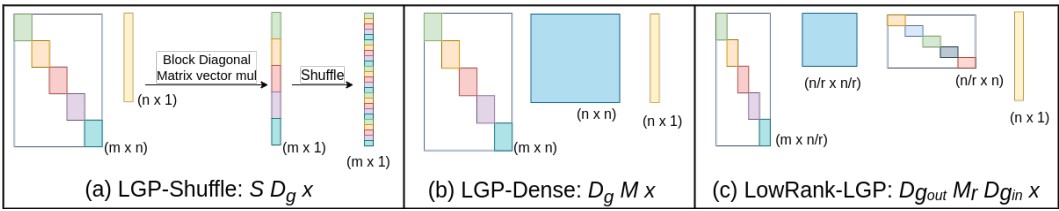

Figure 1: Three different AntMan modules to compress $Ax$.

# 3 ANTMAN DESIGN AND IMPLEMENTATION

AntMan compresses RNN computation by combining benefits of structured sparsity and low rank decomposition. It consists of three components: i) localized group projections that sparsify matrix multiplications using block diagonal matrices, ii) group mixing that exchanges information across different local groups along the sequence of RNN computation, and iii) low rank approximation that uses SVD like decomposition to reduce the rank of the projection. By composing them, we construct a few variations of AntMan compression modules that exhibit varying degree of compression rate and accuracy impact. We also analyze the cost complexity of AntMan modules and discuss efficient implementation on commodity hardware such as CPUs using off-the-shelf BLAS libraries.

## 3.1 LOCALIZED GROUP PROJECTIONS

AntMan reduces the computation and size of RNNs by replacing dense matrix-vector product (MV) with sparse but structured MV. It divides the input and output vectors into $g$ local groups such that the elements of an output group is a weighted linear sum of the elements in the corresponding input group. Since output elements of one group only depend on the input elements of the corresponding group, we call it localized group projections (LGP). Mathematically, we replace the matrix vector product $A\mathbf{x}$ with $D_g\mathbf{x}$, where $D_g$ is a block-diagonal matrix with $g$ blocks.

In an RNN cell computation, the hidden-state vector at time-step $t - 1$ is an input to the MV used to compute the hidden-state vector at time-step $t$. Therefore, using LGP to replace MV in RNN restricts the information flow within a single local group across multiple time steps of the RNN. This restriction reduces the expressibility of the RNN, potentially degrading accuracy. AntMan uses 'group mixing' to address this issue.

## 3.2 GROUP MIXING

To facilitate the information flow across multiple localized groups along RNN sequence computation, AntMan multiplies the output (or input) vector of LGP with a square matrix, which we call mixing matrix. We develop two types of mixing with varying memory and computational complexity — shuffle mix and dense mix — inspired by the shuffling layer (Zhang et al., 2017) used with group convolutions, or 1x1 convolutions used with depth-separable convolutions (Chollet, 2017).

**Shuffle mix:** The shuffle-mix matrix is a permutation matrix, which evenly distributes the elements of the same group across the entire output vector across different groups. Figure 1a shows the effect of shuffle mix following LGP. Mathematically, shuffle mix is equivalent to a transpose operation. If the output vector $\mathbf{v}$ resulting from the block diagonal MV has $m$ elements, we can represent the vector as a matrix $O$ of size $[g, m/g]$, where each row represents an output group computed from the corresponding input group. Shuffle mix simply transforms $\mathbf{v}$ to $O^T$ of shape $[m/g, g]$.

**Dense mix:** This technique uses a dense square matrix for group mixing when the matrix in the MV is non-square. Mathematically, given $A\mathbf{x}$, where size of $A$ is $m$ x $n$, we can decompose it into $MD_g\mathbf{x}$, when $m < n$, or $D_gM\mathbf{x}$, when $n < m$, and $M$ is a dense-mixing matrix of size $m$ x $m$, or $n$ x $n$, respectively. Figure 1b shows an example of dense mix preceding LGP.

Dense mix has added cost of the dense matrix vector multiply compared to shuffle mix (quadratic vs linear). However, unlike shuffle mix that simply permutes the elements of the output vector, dense mix takes a weighted linear combination, making it more general. It helps retain accuracy at the expense of additional computation. When combined with low-rank decomposition discussed next,

dense mix provides high compression while maintaining accuracy, which we elaborate further in evaluation (Table 4).

## 3.3 LOW-RANK DECOMPOSITION

Low-rank decomposition such as SVD approximates a low-rank matrix-vector product $A\mathbf{x}$ as $PQ\mathbf{x}$, where $A$, $P$ and $Q$ are dense with shapes $m$ x $n$, $m$ x $\frac{n}{r}$ and $\frac{n}{r}$ x $n$, respectively, and $\frac{n}{r}$ is the reduced rank. We combine it with LGP by adding LGP-based sparsity on $P$ and $Q$, further reducing computation. This combination is likely to obtain more compression than using either of the techniques alone because structured sparsity and low-rank decomposition operate in a complimentary fashion. In particular, low rank reduces the computation by factorizing $A$ into smaller matrices $P$ and $Q$, while LGP reduces computation by sparsifying these matrices without changing their dimensions.

## 3.4 COMPOSING COMPONENTS INTO ANTMAN COMPRESSION MODULES

Composed from the three components, *LGP*, *group mixing* and *low-rank decomposition*, we construct variations of AntMan compression modules to address varying efficiency and accuracy demand across DL models. Figure 1 shows three of such compression modules: (a) LGP-shuffle — LGP with shuffle mix; (b) LGP-dense — LGP with dense mix; (c) LowRank-LGP — low rank with LGP-dense.

We elaborate the compression modules by taking Figure 1(c), LowRank-LGP, as an example. LowRank-LGP combines structured sparsity with low rank decomposition. First, it decomposes an MV into an SVD-like form, i.e., $A\mathbf{x} \leftarrow PQ\mathbf{x}$, where $A$ is a matrix of size $m$ x $n$, $P$ and $Q$ are decomposed matrices of size $m$ x $\frac{n}{r}$ and $\frac{n}{r}$ x $n$, and $\frac{n}{r}$ represents the reduced rank. Next, we replace $P$ and $Q$ using LGP-Dense, i.e., $A\mathbf{x} \leftarrow D_{g_{out}}M_{out}M_{in}D_{g_{in}}\mathbf{x}$ , where $D_{g_{in}}$ and $D_{g_{out}}$ are block-diagonal matrices of size $m$ x $\frac{n}{r}$ and $\frac{n}{r}$ x $n$, and $g_{in}$ and $g_{out}$ are the number of diagonal blocks. $M_{in}$ and $M_{out}$ are both square matrices of size $\frac{n}{r}$ x $\frac{n}{r}$. It can be further simplified into $A\mathbf{x} \leftarrow D_{g_{out}}M_rD_{g_{in}}\mathbf{x}$, where $M_r = M_{out}M_{in}$. This module, combining all three components of AntMan, exhibits the potential of achieving significantly higher cost reduction than using SVD alone, which we quantify shortly.

## 3.5 COMPUTATION AND MODEL SIZE REDUCTION

| Name | Form of Computation | Computation / Model Size | Cost Reduction |
|---|---|---|---|
| Matrix-Vector | $A\mathbf{x}$ | $mn$ | $1$ |
| SVD | $P_rQ_r\mathbf{x}$ | $\frac{mn}{r} + \frac{nn}{r}$ | $\frac{mr}{m+n}$ |
| LGP-Shuffle | $SD_g\mathbf{x}$ | $\frac{mn}{g}$ | $\mathbf{g}$ |
| LGP-Dense | $MD_g\mathbf{x}$ or $D_gM\mathbf{x}$ | $\frac{mn}{g} + min(m,n)^2$ | $\frac{mn}{\frac{mn}{g}+min(m,n)^2}$ |
| LowRank-LGP | $D_{g_{out}}M_rD_{g_{in}}\mathbf{x}$ | $\frac{mn}{rg_{out}} + \frac{nn}{rg_{in}} + \frac{n^2}{r^2}$ | $\frac{mr^2g_{out}g_{in}}{mg_{in}r+ng_{out}r+ng_{out}g_{in}}$ |

Table 1: Comparison of model computation and size reduction.

Table 1 discusses the reduction in computation and model size over the original $A\mathbf{x}$, where $A$ is a matrix of size $m$ x $n$. The third column reports the total number of multiply-add operations, which is also the size of weight matrix in the case of MV. The final column represents the reduction in computation (that is equal to the reduction in model size) compared to the original MV.

We highlight two key messages: (1) LGP-Dense reduces the total cost by $\approx \frac{max(m,n)}{min(m,n)}$ when $g \gg \frac{max(m,n)}{min(m,n)}$, i.e., the larger difference between $m$ and $n$, the more reduction it gets. (2) When $g_{out}$ and $g_{in}$ are large enough, LowRank-LGP can enable significantly higher cost reduction over SVD, while maintaining the same reduced rank. To see this, let's assume $m = n$, and $g_{out} = g_{in} = g$. In this case, the cost reduction from SVD is $r/2$, while the cost reduction from LowRank-LGP is $\frac{r^2g}{2r+g}$. Now, if $g \geq r$, then the cost reduction is at least $r^2/3$, and it goes up to $r^2$ when $g \gg r$. Therefore, the reduction in computational cost scales as $O(r)$ for SVD, while it scales as $O(r^2)$ for LowRank-LGP assuming $g \geq r$.

As a concrete example, consider a MV of size 1000x400, where the number of parameters and the number of multiply-add (MADD) operations in 400K. Using LGP-Shuffle with $g = 10$, we can reduce both the number of parameters and MADD operations to 40K. Using LGP-Dense with $g = 10$, we can reduce them to 200K (40K from LGP + 160K from dense mix). Using LowRank-LGP with $g = 10$ and $r = 4$, we can reduce the parameters and MADD operations to $\frac{1000*400}{4*10} + \frac{400*400}{4*4} + \frac{400*400}{4*10}$, which is 24K.

### 3.6 EFFICIENT IMPLEMENTATION

We develop efficient implementation of AntMan modules (LGP-Shuffle, LGP-Dense, LowRank-LGP) on CPUs to empower their usage in practice. The implementation consists of three building blocks: i) Regular matrix-vector multiply for dense mix, ii) shuffle-mix multiply and iii) block-diagonal MVs. BLAS libraries such as Intel MKL already provides efficient implementation of matrix-vector multiplication. Shuffle mix is implemented efficiently as a matrix transpose operation as described in Section 3.1. The block-diagonal MV is viewed as multiple smaller MVs, each corresponding to one of the blocks in the block-diagonal matrix. With multicores, each of these blocks can be computed in parallel using OpenMP for parallelization and Intel MKL for MV computation. In summary, AntMan modules can be implemented efficiently on commodity hardware, such as CPUs, conveniently applicable to various devices on cloud and on edge.

## 4 TRAINING ANTMAN USING KNOWLEDGE DISTILLATION

We observe that while training AntMan models directly on target labels alone does not generalize well on test data, using knowledge distillation or teacher-student training helps greatly on retaining accuracy. We use the original uncompressed model as the teacher, and train the compressed model (student) to imitate the output distribution of the teacher, in addition to training on the target labels. We describe how we apply and extend teacher-student training.

**Loss function:** We define the loss function of the compressed model as a weighted combination of *three* losses — the raw loss from the target labels, and the MSE and the KL divergence losses of the student's output distribution with respect to the teacher's corresponding output distribution:

$$\text{Loss}_{total} = C_{target} \times \text{Loss}_{target}(S_o, T_{target}) + C_{mse} \times \text{Mse}(S_o, T_o) + C_{kl} \times \text{KL}(S_o, T_o) \quad (1)$$

where $C_{target}, C_{mse}, C_{kl}$ are the coefficient values corresponding to the target loss, MSE loss and KL divergence loss, respectively. $S_o, T_o$ are the output distributions of student and teacher model, respectively, whereas $T_{target}$ is the target distribution.

**Deciding loss coefficients:** The final performance of the compressed model significantly depends on the values of the loss coefficients, $C_{target}, C_{mse}$ and $C_{kl}$. Searching for appropriate values for these coefficients via grid or random search is time and resource consuming. We develop an efficient method to decide them with the following intuition. The direction of the gradient updates to the model is dictated by the relative magnitudes of the individual losses. If one is significantly smaller than the other, then it would have minimal impact on the direction of the updates during training. Therefore, we want to scale each of the three losses such that the overall magnitude of each of the three terms in Eqn. 1 is roughly the same. To this end, we initially train the compressed model separately using each of the three losses and record the loss values once the model converges. Then we use these values as reference to identify loss coefficients such that each of the three terms is roughly the same. We use these coefficients to train the compressed model to optimize accuracy.

**Effectiveness:** We demonstrate the effectiveness of our approach by training a 10x compressed language model constructed by replacing the LSTMs in Zaremba et al. (2014) with LGP-Shuffle with $g = 10$. For this compressed model, the validation loss values at convergence when training separately with the three individual losses were: target = 4.110, MSE = 0.133 and KL = 0.004.

Table 3 shows the test perplexity values (lower the better) obtained by training the compressed model with varying $C_{mse}$ and $C_{KL}$, while fixing $C_{target} = 1$. Note that the lowest perplexity is achieved by setting coefficient values of $C_{target} = 1$, $C_{mse} = 30$ and $C_{KL} = 1000$. At these values, each term in Eqn. 1 is roughly equal to 4, demonstrating the effectiveness of our method.

Table 3 also shows the benefits of combining *three* losses. Note that when $C_{mse} = 0$, the best achievable perplexity is 121.97. Similarly, when $C_{KL} = 0$, the best achievable perplexity is 75.61. However, combining all three gives the lowest perplexity of 74.69.

| Model | Comp Red. | Test | weights# LSTMs |
|---|---|---|---|
| Zaremba-14 | 1x | 77.551 | 36.00M |
| ISS | 7x | 76.030 | 4.83M |
| ISS | 10x | 78.650 | 3.66M |
| LGP-Shuffle | 10x | 74.693 | 3.60M |
| LGP-Shuffle | 50x | 77.384 | 0.72M |
| LGP-Shuffle | 100x | 78.666 | 0.36M |

Table 2: Computation reduction of models on PTB data and test perplexity values.

| $C_{mse}$ \ $C_{kl}$ | 0 | 1 | 100 | **1000** | 10000 |
|---|---|---|---|---|---|
| 0 | 172.25 | 135.87 | 131.47 | 121.97 | 127.66 |
| 1 | 91.65 | 91.60 | 90.98 | 90.81 | 91.67 |
| **30** | 75.61 | 81.43 | 75.47 | **74.69** | 75.39 |
| 100 | 76.91 | 84.00 | 76.65 | 76.72 | 76.84 |
| 500 | 78.73 | 86.29 | 78.88 | 78.63 | 78.87 |

Table 3: Different choices of coefficients vs test perplexities for student model with 10x computation reduction on the PTB dataset.

## 5 EXPERIMENTS

We evaluate AntMan on three aspects. (1) We use AntMan to obtain order(s) of magnitude computation reduction for language modeling and machine reading comprehension tasks while getting similar accuracy. (2) We use AntMan to construct models with several times fewer parameters than the state-of-art models with the same accuracy targets. (3) Not limited by theoretical speedup, we measure real speedup of AntMan on CPUs and observe super-linear computational efficiency on large RNNs, demonstrating attractive practical value of AntMan on commodity hardware.

### 5.1 COMPUTATION REDUCTION

We evaluate the effectiveness of AntMan on reducing model computation: (1) on Zaremba et al. (2014) model for word level completion task, we obtain 50x reduction without sacrificing any accuracy; (2) on Seo et al. (2016) model for machine reading compression task, we obtain up to 25x reduction with less than 1pt drop on F1 score.

#### 5.1.1 WORD LEVEL COMPLETION

Word level completion task predicts the next word given a partial input sequence.

**Dataset and Model:** We use Penn Tree Bank(PTB) dataset (Marcus et al., 1993) that consists of 929k training words, 73k validation words and 82k test words. As the teacher model, we chose the model in Zaremba et al. (2014) consisting of 2 layered LSTMs each with hidden dimension of 1500. For the student model, we replace all the MVs in the LSTMs with LGP-Shuffle, and use $g = 10$ to $g = 100$ groups. We do not use any low-rank decomposition for this model.

**Results:** Table 2 shows the perplexity values of the compressed models for different levels of computation reductions. Matching the perplexity of the original model, AntMan ($g = 50$) achieves 50x computation reduction. With $g = 10$, AntMan achieves 10x computation reduction while 3pt better perplexity. With $g = 100$, AntMan achieves 100x computation reduction with only 1pt loss in perplexity. In addition, comparing with the state-of-art compressed model in Wen et al. (2017) using ISS, AntMan reduces computations *further* by 5-10x under comparable test perplexities.

#### 5.1.2 MACHINE READING COMPREHENSION (MRC)

MRC tasks have gained significant popularity in last few years within NLP and computer vision communities. The models answer a query about a given context paragraph, evaluated based on exact match (EM) and F1 score (higher the better).

**Dataset:** We use Stanford Question Answering Dataset (SQuAD) (Rajpurkar et al., 2016), which consists of a large set of Wikipedia articles and more than 100,000 questions. The answer to every question is always a small excerpt of the article. **Teacher Model:** We chose our teacher model as the BiDirectional Attention Flow Model (BiDAF) (Seo et al., 2016), which is a hierarchical multi-stage model with 6 layers. We focus on compressing the layers having RNNs, which are also the most computationally expensive ones. Specifically, the modeling layer uses 2 layers of bi-directional LSTMs, denoted by ModFwd1, ModBwd1, ModFwd2, ModBwd2, while the output layer has a single bi-directional LSTM, denoted by OutFwd, OutBwd.

**Compressed Models:** We created three compressed models using AntMan with different levels of compression to replace the LSTMs in the BiDAF model: i) LGP-Shuffle ($g_{im} = 10, g_{hm} = 4$),

| Description | EM | F1 | ModFwd1 | ModBwd1 | ModFwd2 | ModBwd2 | OutFwd | OutBwd | weight# |
|---|---|---|---|---|---|---|---|---|---|
| Expert | 67.9 | 77.3 | 1x | 1x | 1x | 1x | 1x | 1x | 2.69M |
| ISS | 65.29 | 75.47 | 1.95x | 2.26x | 6.14x | 4.34x | 5.87x | 8.85x | 1.03M |
| **LGP-Shuffle** | 65.41 | 75.87 | 9.09x | 9.09x | 6.66x | 6.66x | 9.09x | 9.09x | 0.78M |
| **LowRank-LGP 1** | 66.06 | 76.73 | 12.5x | 12.5x | 9.09x | 9.09x | 16.66x | 16.66x | 0.69M |
| **LowRank-LGP 2** | 65.86 | 76.6 | 20.42x | 20.42x | 17.7x | 17.7x | 25.39x | 25.39x | 0.56M |

Table 4: Comparision of computation reduction between AntMan and ISS for BiDAF

ii) LowRank-LGP 1 ($g_{im} = 10, g_{hm} = 5, r_{im} = 4, r_{hm} = 2$), and iii) LowRank-LGP 2 ($g_{im} = 5, g_{hm} = 5, r_{im} = 8, r_{hm} = 4$). Here, $g_{im}$ and $g_{hm}$ refers to the number of groups, and $r_{im}$ and $r_{hm}$ refers to the low-rank reduction factors for input and hidden MVs of the LSTMs, respectively. The computation reduction for each LSTM is shown in Table 4.

**Results:** Table 4 shows that both LGP-Shuffle and LowRank-LGP achieve significant computation reduction over the original; their reduction is much higher than the existing work ISS (Wen et al., 2017) with better EM and F1 scores. ISS compresses an RNN by reducing hidden dimension. The amount of computation per LSTM step for ISS is proportional to $(i + h/r) * (h/r)$, where $i$ is the input dimension, $h$ is the hidden dimension, and $1/r$ is fraction of hidden dimension removed by ISS. When $i \gg h$, the compression is proportional to $r$. In BiDAF, $i \gg h$ in the first modeling layers (800 vs 100). Therefore, compression in these layers is proportional to the reduction in the hidden dimension. However, $h = 100$ is already very small. By reducing it further, ISS experiences near 2pt drop in F1 score with less than 2.5x compression on the first modeling layers.

LGP-Shuffle uses structured sparsity to compress both the input and hidden MVs without reducing hidden dimension. For a comparable EM and F1 scores to ISS, LGP-shuffle achieves significantly higher reduction on the first modeling layers, while doing modestly better on all other layers. LowRank-LGP improves further upon LGP-Shuffle, increasing accuracy by leveraging dense mix to enrich the connection among multiple localized groups, and reducing computation by combining low-rank decomposition. It achieves significantly higher computation reduction across all layers than both ISS and LGP-Shuffle, while achieving nearly 1pt higher F1 scores.

## 5.2 OPTIMIZED MODEL SIZE FOR DIFFERENT ACCURACY TARGETS

Different applications have various accuracy requirements while the devices they are running on also impose different constraints on the model size. For given accuracy targets, smaller models are desired; and for given model sizes, higher accuracy is desired. We show that AntMan improves the Pareto curve of model size against accuracy, providing more compressed models with the same accuracy targets of several recent models at word level completion task.

**Teacher Model:** We use the state-of-art language model as of Sept, 2018, AWD-LSTM (Merity et al., 2017), consisting of 3-layer LSTMs with 1150 hidden units and an embedding size of 400.

**Compressed Models:** Our compressed models replace all the MVs in the LSTMs of AWD-LSTM with AntMan (LGP-Shuffle with $g = 5$ to $g = 50$ groups).

**Results:** Figure 2 compares AntMan with other models. LGP-Shuffle ($g = 5$) achieves perplexity of 63 with 5x fewer LSTM parameters and 3x fewer total parameters than NAS-Cell (Zoph & Le, 2016), the state-of-art model obtaining this range of accuracy. LGP-Shuffle ($g = 10$) achieves perplexity of 66 with 10x fewer LSTM parameters and 4x fewer total parameters than Var-RHN (Zilly et al., 2016), and LGP-Shuffle ($g = 50$) achieves perplexity of 74 with 50x fewer LSTM parameters and 5x fewer total parameters than Var-LSTM-avg1 (Inan et al., 2016). These results notably improve the Pareto curve of the task by reducing model sizes against different accuracy targets.[1]

---

[1] We did not aim to reproduce the state-of-art perplexity (57.3px at 24M parameters) of AWD-LSTM model. AWD-LSTM uses various regularization techniques, each with its own set of hyper-parameters, requiring extensive hyper-parameter tuning to reach its state-of-art perplexity. The AntMan results presented in Figure 2 was achieved without any regularization. Trying to match AWD-LSTM perplexity using AntMan with regularization could be an exercise in large scale hyper-parameter tuning, which is beyond the scope of this paper.

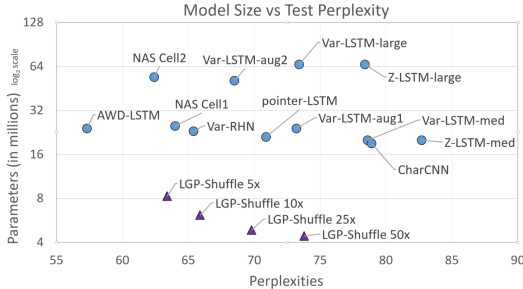

| Input & Hidden Dim | Memory (MB) | Actual vs Theoretical Speedup | | | |
|---|---|---|---|---|---|
| | | LGP-Shuffle | | LowRank-LGP | |
| | | 2x | 10x | 2.66x | 8x |
| 100 | 0.32 | 1.10x | 1.16x | 1.10x | 0.08x |
| 400 | 5.12 | 1.89x | 6.39x | 2.32x | 4.70x |
| 800 | 20.48 | 2.00x | 8.66x | 2.78x | 6.50x |
| 1200 | 46.08 | 4.80x | 24.02x | 6.50x | 20.00x |
| 1600 | 81.92 | 5.40x | 30.20x | 7.42x | 23.80x |

Figure 2: Comparing the number of model parameters vs perplexity of AntMan based models with various other language models published in the last four years, extracted from Table1 in (Merity et al., 2017). The AntMan based models (LGP-Shuffle) are shown as purple triangles.

Table 5: Measured speedup on CPU using LGP-Shuffle and LowRank-LGP compared to the theoretical speedup for various input and hidden dimension. For LGP-Shuffle, we use $g = 2$ and $g = 10$ to get a theoretical speedup of 2x and 10x. For LowRank-LGP, we use $g = 2$ and $r = 2$, and $g = 10$, and $r = 2$ to get a speedup of 2.66x and 8x, respectively.

### 5.3 Theoretical vs Actual Speedup

By using efficient implementation of AntMan described in Section 3.6, we turn the theoretical speedup (computation reduction) to actual speedup (execution time reduction) in practice. Furthermore, we show that the actual speedup can be significantly higher than the theoretical speedup for large problem sizes. The result of our evaluation is shown in Table 5.

**Problem Configuration:** We measure the execution time of LSTMs with and without AntMan varying input and hidden dimensions from 100 to 1600. We use a batch size of 1, which is common in serving scenarios, and a sequence length of 100.

**LSTM Implementation:** We use an efficient implementation as discussed in Elsen: Fuse 4 input MVs across all time steps into a single large matrix multiplication, and fuse 4 hidden MVs within each time step.

**Platform:** The experiments are run on a single core of Intel CPU E5-2650 v4 @ 2.20GHz. We use just a single core for two reasons: i) to emulate the limited resource availability in many use cases such as laptops and smart phones, ii) performance of multi-core RNN is highly implementation dependent (Zhang et al.) even for regular RNNs and therefore is difficult to make apple-to-apple comparison. We use Intel MKL library for GEMM implementation.

**Discussion:** Table 5 shows that, for very small problem size, AntMan offers no speedup regardless of the reduction in the computation. This is expected as GEMM performance gets worse as the problem size decreases (Zhang et al.; Rajbhandari et al., 2017). However, as the problem size is already very small, memory reduction or performance improvement is less crucial for such problems. For medium sized problems, AntMan offers good actual speedup compared to the theoretical speedup. Notice that unlike unstructured sparsity, where significant levels of sparsity is necessary to see actual performance improvement, with AntMan, even a modest 50% sparsity or 2x computation reduction results in significant performance gain at problem size 400 and 800. Furthermore, for large problem sizes the actual speedup is significantly larger than the theoretical speedup. At problem size of 1200 and 1600, the weight matrices in the LSTM are too large to fit in L3 cache (30 MB in this case), thus spilling into memory. These LSTMs have much lower efficiency as the memory bandwidth of a CPU is much lower than the L3 cache bandwidth. By reducing the memory footprint, AntMan-based LSTM fits in L3 cache, leading to an actual speed up that is considerably higher than the theoretical speedup. These results demonstrate attractive practical value of AntMan on commodity hardware.

## 6 Conclusion

We develop AntMan, combining structured sparsity and low-rank decomposition, to reduce the computation, size and execution time of RNN models by order(s) of magnitude while achieving similar accuracy. We hope its compression efficiency and effectiveness would help unblock and enable many great RNN-based models deployed in practice.

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

| Compute Reduction | Small RNN | SVD RNN | AntMan |
|---|---|---|---|
| 10x | 80.06 | 78.63 | 74.6 |
| 50x | 99.96 | 81.4 | 77.4 |
| 100x | 129.3 | 88.59 | 78.6 |

Table 6: Computation reduction of models on PTB data and test perplexity values.

| Theoretical Compression | Actual Performance Gain | |
|---|---|---|
| | Pruning | AntMan |
| 10x | 4x | 30x |

Table 7: Theoretical vs Actual Performance gain on PTB using unstructured Pruning vs AntMan

APPENDIX

A. CHOICE OF BASELINE

We considered several compression techniques to identify strong baselines to compare with AntMan. Eventually, we chose ISS as the main baseline in the paper, because for RNNs ISS satisfies two important criteria that we believe are most critical for model compression techniques, i) Good theoretical reduction in compute and memory achievable without sacrificing accuracy, ii) Good computation efficiency of the compressed model to fully exploit the theoretical reduction in computation. Most compression techniques do not satisfy both..

We discuss and compare AntMan with several compression techniques as below.

**Quantization**: 16 and 8-bit quantization (original 32-bit) can be supported fairly easily on commodity hardware, resulting in a maximum compression of 4x. Even more aggressive quantization (e.g., 2-7 bit) hardly provides additional computational benefit because commodity hardware does not support those in their instruction set, while 1-bit quantization does not offer comparable accuracy.

In comparison, we demonstrate that AntMan achieves up to 100x reduction in computation without loss in accuracy. Moreover, quantization can be applied to AntMan to further reduce the computation, and vice versa, as quantization and AntMan are complementary techniques.

**Pruning:** Pruning can be used to generate both unstructured and structured sparsity. The former is not computationally efficient while the latter requires specialized implementation for efficient execution.

While we did not present pruning results in the paper, we did try out techniques on both PTB and BiDAF models to generate random sparsity as well as blocked sparsity. In both cases, we were able to get more that 10x reduction in computation even in the absence of Knowledge distillation. Therefore pruning provides excellent computation reduction.

However, as discussed in the paper, those theoretical computational reductions cannot be efficiently converted into practical performance gains: Unstructured sparsity resulting from pruning suffers from poor computation efficiency; a 10x theoretical reduction leads to less than 4x improvement in performance while AntMan achieves 30x performance gain with 10x reduction for PTB like models. (Table 7)

It is possible to achieve structured sparsity such as block sparsity through pruning. However, structured sparsity requires implementing specialized kernels to take advantage of the computation reduction. Its efficiency greatly depends on the implementation, and in general is far from the theoretical computation reduction.

On the contrary both ISS and AntMan achieve good computation reduction, and can be efficiently executed using readily available BLAS libraries such as Intel MKL resulting in super linear speedups as shown in the paper.

**Direct Design**: We compared AntMan with smaller RNN models (with smaller hidden dimension) trained using the larger teacher model. Our results show that for the same level of compression AntMan achieves much higher accuracy. (Table 6)

**SVD RNN**:We constructed compressed models by replacing matrix-multiplication with SVD of various rank, and trained the SVD based models using knowledge distillation. Once again, we find

that for the same level of compression, AntMan achieves much higher accuracy than SVD. (Table 6)

**Block Tensor Decomposition (BTD)** : BTD is designed to compress RNNs whose inputs are produced by convolution based models, and contain certain redundancies. AntMan, on the other hand, is generic to all RNN based models. Also, BTD is designed to compress only the input vector and not the hidden vectors. This hinders the performance of BTD over a range of RNNs, where the hidden vectors are also large.

## B. ISS vs AntMan without Knowledge Distillation

| Description | EM | F1 | ModFwd1 | ModBwd1 | ModFwd2 | ModBwd2 | OutFwd | OutBwd | weight# |
|---|---|---|---|---|---|---|---|---|---|
| Expert | 67.9 | 77.3 | 1x | 1x | 1x | 1x | 1x | 1x | 2.69M |
| ISS | 65.29 | 75.47 | 1.95x | 2.26x | 6.14x | 4.34x | 5.87x | 8.85x | 1.03M |
| **LowRank-LGP** | 66.07 | 76.11 | 8.6x | 8.6x | 7.5x | 7.5x | 11.2x | 11.2x | 0.60M |

Table 8: Comparison of computation reduction between AntMan and ISS for BiDAF without Knowledge Distillation

Here, we compare the performance of AntMan with ISS, without using any knowledge distillation. Please note that knowledge distillation is part of the training process for AntMan, but it is not for ISS. Nevertheless, it is interesting to see how AntMan performs in the absence of a teacher.

When trained without knowledge distillation, our experiments show that AntMan and ISS have complimentary strengths. On the PTB dataset, with a 10x compute reduction, AntMan does not generalize well without a teacher, while ISS incurs less than 1pt loss in perplexity compared to the original model. This is demonstrated by the first row and column in Table 3, and the third row in Table 2. On the contrary, for the BiDAF, AntMan incurs less than 1pt reduction in F1 score for nearly 10x compute reduction[2], while ISS incurs nearly 2pt reduction in F1 score with less than 5x compute reduction on average. This is shown in Table 8.

AntMan can successfully compress BiDAF, while ISS fails because ISS compresses an LSTM by effectively reducing its hidden dimension, while AntMan preserves the hidden dimension size. The LSTMs in the BiDAF model have large input dimensions making them computationally expensive, but they have very small hidden dimensions. Therefore, reducing the already small hidden dimension results in significant loss of accuracy. On the contrary, the PTB model has large input as well as hidden dimensions, allowing ISS to work effectively.

## C. Additional Experiment details

**Optimizers** All the models in the evaluation section were trained using ADAM optimizer.

**Hyperparameters** For both PTB and SQUAD, all the hyper-parameters for training the respective models were set as default from their standard and recommended implementations on github (PTB [3], BiDAF[4]).

---

[2]LowRank LGP with $g_{im} = 5, g_{hm} = 5, r_{im} = 4, r_{hm} = 2$

[3]https://github.com/tensorflow/models/tree/master/tutorials/rnn/ptb

[4]https://github.com/allenai/bi-att-flow

