# OpenReview forum: "AntMan: Sparse Low-Rank Compression To Accelerate RNN Inference"
_ICLR.cc/2019/Conference_

### Official Review · AnonReviewer2 · 2018-10-29
**Nice idea but the experiments are quite insufficient**

**Rating:** 5
**Confidence:** 4

**Review:**

This paper presents a network compression method based on block-diagonal sparse structure for RNN. Two kinds of group mixing methods are discussed. Experiments on PTB and SQUAD have shown its superiority over ISS.
The idea present is interesting, and this paper is easy to follow. However, this paper can be improved from the following perspectives.
1.	The method of balancing the quantity of different parts in knowledge distillation is trivial. It is quite general trick.
2.	Details of experimental setup were unclear. For example, the optimization method used, the block size, and the hyper-parameters were unclear. In addition, it is also unclear how the block diagonal structure was used for the input-to-hidden weight matrix only or all weights.
3.	In addition, the proposed method was compared with ISS only. Since there are many methods of compressing RNNs, comparison with other competitors (e.g., those presented in Related work) are necessary.  Moreover, more experiments with other tasks in addition to NLP will be better.
4.	In Table 2, the comparison with ISS seems be unfair. The proposed methods, i.e., LGP-shuffle was obtained based on the distillation. However, ISS was trained without distillation. From Table 3, when Cmse and Ckl were set to zero, the result was much worse. The reviewer was wondering that how does ISS with distillation perform.

---

> ### Author Response · Authors · 2018-11-21
> **Response to Comment 4**
>
> 4. Unfair comparison with ISS
>
> While we agree that ISS with KD would be interesting to try out, we do consider our current comparison fair as we are comparing with the full set of techniques described in the ISS paper. The authors of the ISS paper do not claim that their techniques should be used together with Knowledge Distillation. On the contrary, KD is part of our training process. Therefore, we are comparing the entirety of our techniques to the entirety of the ISS techniques.
>
> Training with ISS would also require finding solutions to additional problems that are not described in the ISS paper. For example, ISS induces sparsity using a group lasso term in the loss function. As KD also introduces additional teacher loss to the loss function, it becomes necessary to balance the different loss terms that are targeted at achieving different goals (sparsity vs accuracy). It is not clear how these terms can be balanced systematically. Further, ISS requires tuning various hyper-parameters such as dropout, weight decay, and learning rate. In the presence of additional terms the loss function, it becomes necessary to do a full hyper-parameter sweep to identify the best set of parameters.
>
> We like the reviewer's suggestion on using ISS with KD and are working actively on getting these results, as our end goal is to find the best compression techniques for our production models. However, we consider it to be non-trivial, and well beyond what is described in the ISS paper.
>
> We also appreciate that the reviewer took an in-depth look at the results to point out that in the absence of KD, AntMan does worse than ISS for PTB. While this is true for PTB, we also want to point out that this is not always the case. For BiDAF, AntMan has better F1 score than ISS for the same level of compression in the absence of KD. For example, using LowRank LGP, in the absence of KD, AntMan can achieve average compute reduction of 8x for BiDAF while achieving F1 score over 76. On the other hand for F1 scores less than 76, ISS provides a compression of less than 5x on average. We have added these results in Appendix B.

---

> ### Author Response · Authors · 2018-11-21
> **Response to Comment 3**
>
> 3. Choice of baseline
>
> We understand the reviewer's concern regarding the choice of baseline. In fact, we considered several compression techniques to identify the strongest baselines to compare with AntMan. Eventually, we chose ISS as the baseline, because for RNNs ISS satisfies two important criteria that we believe are most critical for model compression techniques, i) Good theoretical reduction in compute and memory achievable without sacrificing accuracy, ii) Good computation efficiency of the compressed model to fully exploit the theoretical reduction in computation.  Most compression techniques do not satisfy both of these criteria.
>
> Based on the reviewer's feedback, we discuss various compression techniques in regards to the aforementioned criteria and how they compare to AntMan. We also provide experimental results when applicable. We have added this discussion and results to the appendix A of the paper.
>
> a. Quantization: 16 and 8-bit quantization (original 32-bit) can be supported fairly easily on commodity hardware, resulting in a maximum compression of 4x. Even more aggressive quantization (e.g., 2-7 bit) hardly provides additional computational benefit because commodity hardware does not support those in their instruction set, while 1-bit quantization does not offer comparable accuracy.
>
> In comparison, we demonstrate that AntMan achieves up to 100x reduction in computation without loss in accuracy.  Moreover, quantization can be applied to AntMan to further reduce the computation, and vice versa, as quantization and AntMan are complementary techniques.
>
> b. Pruning: Unstructured sparsity resulting from most pruning techniques is not computationally efficient. For both PTB and BiDAF, our experiments show that pruning can achieve over 10x in computation reduction without even using knowledge distillation. However, due to poor computational efficiency, 10x theoretical reduction translates to less than 4x reduction in wall-clock time, while 10x reduction in Ant-Man translates up to 30x reduction in wall-clock time as demonstrated in the paper. Due to this inefficiency in computation, we do not consider pruning as a competitive baseline to compare against. (Table 2)
>
> Pruning can also be used to produce structured sparsity such as blocked sparsity. However, structured sparsity requires implementing specialized kernels to take advantage of the computation reduction. Its efficiency greatly depends on the implementation, and in general is far from the theoretical computation reduction.
>
> c. Direct Design: Reviewer 2 suggested comparing with AntMan with smaller RNN models trained using the larger teacher model. Our results show that for the same level of compression AntMan achieves much higher accuracy.  (Table 3)
>
> d. SVD : We constructed compressed models by replacing matrix-multiplication with SVD, and trained the SVD based models using knowledge distillation. Once again, we find that for the same level of compression, AntMan achieves much higher accuracy than SVD. (Table 3)
>
> e. Block Tensor Decomposition (BTD): BTD is designed to compress RNNs whose inputs are produced by convolution based models, and contain certain redundancies. AntMan, on the other hand, is generic to all RNN based models. Also, BTD is designed to compress only the input vector and not the hidden vectors. This hinders the performance of BTD over a range of RNNs, where the hidden vectors are also large and hence we didn’t consider BTD as a strong baseline to compare against.
>
> In summary, we considered several compression techniques when choosing our baseline. We found that most techniques either did not achieve good computation reduction with comparable accuracy or were computationally inefficient to execute.  ISS is a strong baseline, providing both good theoretical reduction and good computational efficiency, which we chose to compare with thoroughly.
>
>
> Table 2: Theoretical vs Actual Performance gain on PTB using unstructured Pruning vs AntMan
>
> Theoretical Compression         Actual Performance Gain
>                                                       Pruning              AntMan
> 10x                                                    4x                         30x
>
> Table 3: Test Perplexity for PTB model for various levels of computation compression using different compression techniques
>
> Compute Reduction        Small RNN Perplexity         SVD RNN Perplexity       AntMan Perplexity
> 10x                                               80.06                                     78.63                                74.6
> 50x                                               99.96                                     81.4                                  77.4
> 100x                                           129.3                                       88.59                                78.6

---

> ### Author Response · Authors · 2018-11-21
> **Response to Comment 2**
>
> 2. Missing experimental details.
>
> We added the missing details in the paper (Appendix C)
>    a. Optimization method used: ADAM
>    b. Block Sizes are described in 5.1.1 and 5.1.2
>    c. Block diagonal structures were used in both input and hidden weights.
>    d. We did not alter any other hyper-parameters for the PTB and BiDAF models. The full set of
>         hyper-parameters can be found here (
>                 PTB: https://github.com/tensorflow/models/tree/master/tutorials/rnn/ptb
>                 BIDAF: https://github.com/allenai/bi-att-flow
>    e. We will also release the codes of our implementations once the paper is accepted so all the
>        results are easily reproducible.

---

> ### Author Response · Authors · 2018-11-21
> **Response to Comment 1**
>
> 1. Balancing different parts of Knowledge Distillation is trivial.
>
> We find the two techniques ((a) using MSE, KL divergence and label loss together, (b) efficient hyperparameter search) quite helpful when applying KD. They are intuitive but we did not find any related work talking about them. Therefore, we hope to share the good practice with the readers of the paper. If they are discussed in any related work (we are not aware), kindly point out and we will be glad to cite and adjust the claims accordingly.

---

> ### Author Response · Authors · 2018-11-21
> **Response to Review**
>
> Thank you for the constructive feedback.   We respond to the 4 comments separately, as OpenReview does not allow us to post a single long response.

---

### Official Review · AnonReviewer3 · 2018-11-02
**Good design**

**Rating:** 5
**Confidence:** 2

**Review:**

This paper proposed to use sparse low-ranking compression modules to reduce both computation and memory complexity of RNN models. And the model is trained using knowledge distillation.
clarity:
I think Fig1a can be improved. Initially I don't understand how the shuffle part works. It will be more clear if the mx1 vectors have the same length and the two (m x1) labels are in the same height.
originality:
The method is quite interesting and should be interesting to many people.
pros:
1) The method reduces computation and memory complexity at the same time.
2) The result looks impressive.
cons:
1) Is the training of AntMan models done on GPU or CPU? How is the training time. It seems efficient implementation of the model on GPU can be challenging.
2) It seems the modules can be used to replace any dense matrix in the neural networks. I'm not sure why it is applied on RNN only.
3) I think another baseline is needed for comparison, a directly designed small RNN model trained using knowledge distillation. In this way, we can see if the sparse low-rank compression provides new values.

---

> ### Author Response · Authors · 2018-11-21
> **Response to Review**
>
> Thank you for the constructive feedback.
>
> 1: Is the training done on CPU or GPUs? How is the training time?
>
> CPU or GPU: While we trained the AntMan models on GPU, our tensorflow implementation is architecture agnostic. It can be trained on either.
>
> Training time and efficient implementation: AntMan implementation is in fact extremely simple. Each AntMan based RNN module can be implemented with less than a 100 lines of tensorflow code, and efficiently trained on either CPU or GPU.
>
> We simply replace the matrix multiplications in the RNN with AntMan modules which are themselves composed of smaller matrix-multiplications. As the basic building block is still a matrix-vector multiplication, they can be computed efficiently on both CPU and GPU.
>
> PTB takes less than 6 hours to train on a single Titan XP GPU, while the BIDAF model takes about 20-40 hours to train. The training time of the compressed model is comparable as of the original model.
>
> 2: Why is AntMan applied just to RNNs?
>
> It is definitely true that AntMan can be applied to any Matrix-vector product in a neural network. We do not claim that AntMan only works for RNN. We focus on RNNs because matrix-vector multiplies are the primary source of performance bottleneck in RNNs. We plan to try out AntMan on other networks as part of our future work.
>
> 3: Another baseline is needed for comparison. The reviewer suggests a directly designed small
> RNN trained using Knowledge distillation.
>
> 	We understand the reviewer's concern regarding the choice of baseline. In fact, we considered several compression techniques to identify the strongest baselines to compare with AntMan. Eventually, we chose ISS as the baseline, because for RNNs ISS satisfies two important criteria that we believe are most critical for model compression techniques, i) Good theoretical reduction in compute and memory, ii) Good computation efficiency of the compressed model that results in an actual reduction in the wall-clock time.  Most compression techniques do not satisfy both of these criteria.
>
> However, we do think that alternate baseline suggested by the reviewer makes good sense. We have run the suggested experiments and added the results comparing AntMan with directly designed small RNN models in the appendix. To summarize, we find that for a given level of compression, directly designed small RNN model trained with the original teacher model has much higher perplexity than AntMan compressed model trained with the same teacher, demonstrating that AntMan indeed provides new value. The results are presented in the table below. We have also added them to the appendix of our paper.
>
> Table 1: Test Perplexity for PTB model for various levels of computation compression
>
> Compute Reduction   Small RNN Perplexity   AntMan Perplexity
> 10x                                        80.06                                        74.6
> 50x                                       99.96                                        77.4
> 100x                                     129.3                                        78.6

---

### Official Review · AnonReviewer1 · 2018-11-03
**Good paper, but needs stronger baseline.**

**Rating:** 6
**Confidence:** 5

**Review:**

Model Compression is used to reduce the computational and memory complexity of DL models without significantly affecting accuracy. Existing works focused on pruning and regularization based approaches where as this paper explores structured sparsity on RNNs, using predefined compact structures.

They replace matrix-vector multiplications which is the building computational block part of RNNs, with localized group projections (LGP). where LGP divides the input and output vectors into groups where the elements of the output group is computed as a linear combination of those from the corresponding input group. Moreover, they use a permutation matrix or a dense-square matrix to combine outputs across groups. They also combine LGP with low-rank matrix decomposition in order to further reduce the computations.

Strong points:

Paper shows how combining the SVD and LGP can reduce computation. In particular in matrix-vector multiplications Ax, low rank reduces the computation by factorizing A into smaller matrices P and Q, while LGP reduces computation by sparsifying these matrices without changing their dimensions.

The paper discussed that their model target labels alone does not generalize well on test data and they showed teacher-student training helps greatly on retaining accuracy. They use the original uncompressed model as the teacher, and train the compressed model(student) to imitate the output distribution of the teacher, in addition to training on the target labels.

Paper is well written and easy to follow.

This paper would be much stronger if it compared against quantization and latest pruning techniques.

This paper replace matrix-vector multiplications with the lowRank-LGP, but they only consider RNN networks. I am wondering how it affects other models given the fact that matrix-vector multiplications is the core of many deep learning models. It is not clear why their approach should only work for RNNs.

Table 1 shows the reduction in computation and model size over the original matrix-vector multiplications Ax. I think in this analysis the computation of the those approaches are neglected. For example running the SVD alone on A (n by m matrix) takes O(m^2 n+n^3). That is true that if P and Q are given, then the cost would be n(m+n)/r. However, finding P and Q takes O(m^2 n+n^3) that could be very expensive when matrices are large.

Table 2 only shows the LGB-shuffle resuts. What about the combined SVD and LGP? Similarly in Table 4, what is the performance of the LGB-Dense?

---

> ### Author Response · Authors · 2018-11-21
> **Response to Review**
>
> Thank you for the constructive feedback.
>
> 1. Comparing against quantization and pruning
>
> We understand the reviewer's concern on the choice of baseline, specifically the lack of comparison with quantization and pruning. In fact, we considered several compression techniques including quantization and pruning to identify the strongest baselines to compare with AntMan. Eventually, we chose ISS as the baseline, because for RNNs ISS satisfies two important criteria that we believe are most critical for model compression techniques, i) Good theoretical reduction in compute and memory, ii) Good computation efficiency of the compressed model that results in an actual reduction in the wall-clock time. Most model compression techniques lack one of these criteria.
>
> While Quantization and pruning are effective techniques for model compression, the former provides limited compression, while the latter is not computationally efficient.  More specifically, we did not consider quantization and pruning as strong baselines for the following reasons.
>
> 1a. Quantization:  Even without running any experiments, we already know the maximum model compression that is achievable using quantization. Commodity hardware can support computation on up to 8-bit integers.   Even more aggressive quantization (e.g., 2-7 bit) hardly provides additional computational benefit because commodity hardware does not support those in their instruction set, while 1-bit quantization does not offer comparable accuracy. So theoretically, the maximum level of compression achievable through quantization alone on commodity hardware is from 32-bit to 8-bit resulting in a maximum compression of  4x, while AntMan can achieve up to 100x with no accuracy loss.
>
> Furthermore, quantization and AntMan are complimentary techniques, which can be used together on a model.
>
> 1b. Pruning: It is difficult to translate the computation reduction from pruning based techniques into actual performance gains.
>
> While we did not present pruning results in the paper, we did try out techniques on both PTB and BiDAF models to generate random sparsity as well as blocked sparsity. In both cases, we were able to get more than 10x reduction in computation even in the absence of Knowledge distillation. Therefore pruning provides excellent computation reduction.
>
> However, as discussed in the paper, those theoretical computational reductions cannot be efficiently converted into practical performance gains: Unstructured sparsity resulting from pruning suffers from poor computation efficiency; a 10x theoretical reduction leads to less than 4x improvement in performance while AntMan achieves 30x performance gain with 10x computation reduction for PTB like models.
>
> It is possible to achieve structured sparsity such as block sparsity through pruning. However, structured sparsity requires implementing specialized kernels to take advantage of the computation reduction.
>
> Due to these reasons, we felt that Pruning techniques are weaker baselines to compare against. On the contrary, both ISS and AntMan achieve good computation reduction and can be efficiently executed using readily available BLAS libraries such as Intel MKL resulting in superlinear speedups. This is why we chose ISS as a primary baseline. We have added a section in the appendix discussing the limitation of various compression techniques including quantization and pruning in comparison to AntMan.
>
>
> 2. Why just RNNs?
>
> It is definitely true that AntMan can be applied to any Matrix-vector product in a neural network. We do not claim that AntMan only works for RNNs. We focus on RNNs because matrix-vector multiplies are the primary source of performance bottleneck in RNNs. We plan to try out AntMan on other networks as part of our future work.
>
>
> 3. Cost of Finding P and Q?
>
> The cost of finding P and Q can be neglected because we do not explicitly calculate them. Lets use SVD (PQ = A) for clarification. Instead of first training to find the weights of matrix A, and then decomposing into P and Q, we replace A as PQ directly in the neural network, allowing the back-propagation to find the appropriate values of P and Q that minimizes the loss function.
>
> 4. What about combining SVD and LGP in Table 2? What about LGP Dense?
> 	In Table 2 we presented only those results that yielded maximum computation reduction with minimum loss in accuracy. For PTB, LowRank LGP incurs more accuracy loss than just LGP. For example, for 10x compression, LGP achieves a perplexity of 74.69 while the best LowRank LGP achieves 75.99.

---

### Meta-Review · Area_Chair1 · 2018-12-16
**Area chair recommendation**

**Confidence:** 5
**Recommendation:** Reject

**Metareview:**

The submission proposes a method that combines sparsification and low rank projections to compress a neural network.  This is in line with nearly all state-of-the-art methods.  The specific combination proposed in this instance are SVD for low-rank and localized group projections (LGP) for sparsity.

The main concern about the paper is the lack of stronger comparison to sota compression techniques.  The authors justify their choice in the rebuttal, but ultimately only compare to relatively straightforward baselines.  The additional comparison with e.g. Table 6 of the appendix does not give sufficient information to replicate or to know how the reduction in parameters was achieved.

The scores for this paper were  borderline, and the reviewers were largely in consensus with their scores and the points raised in the reviews.  Given the highly selective nature of ICLR, the overall evaluations and remaining questions about the paper and comparison to baselines indicates that it does not pass the threshold for acceptance.